# Polyacrylonitrile Fibers with a Gradient Silica Distribution as Precursors of Carbon-Silicon-Carbide Fibers

**DOI:** 10.3390/polym15112579

**Published:** 2023-06-05

**Authors:** Lydia A. Varfolomeeva, Ivan Yu. Skvortsov, Ivan S. Levin, Georgiy A. Shandryuk, Timofey D. Patsaev, Valery G. Kulichikhin

**Affiliations:** 1A. V. Topchiev Institute of Petrochemical Synthesis of Russian Academy of Sciences, Leninsky Av. 29, 119991 Moscow, Russia; varfolomeeva.lidia@mail.ru (L.A.V.); levin@ips.ac.ru (I.S.L.); gosha@ips.ac.ru (G.A.S.); 2National Research Center “Kurchatov Institute”, 1, Akademika Kurchatova pl., 123182 Moscow, Russia; timpatsaev@mail.ru

**Keywords:** polyacrylonitrile, fiber spinning, silica nanoparticles, carbon fibers with silicon carbide, wet spinning, mechanotropic spinning, tetraethoxysilane, sol-gel method, precursor fibers, fiber morphology

## Abstract

This study presents preparing and characterization of polyacrylonitrile (PAN) fibers containing various content of tetraethoxysilane (TEOS) incorporated via mutual spinning solution or emulsion using wet and mechanotropic spinning methods. It was shown that the presence of TEOS in dopes does not affect their rheological properties. The coagulation kinetics of complex PAN solution was investigated by optical methods on the solution drop. It was shown that during the interdiffusion process phase separation occurs and TEOS droplets form and move in the middle of the dope’s drop. Mechanotropic spinning induces the TEOS droplets to move to the fiber periphery. The morphology and structure of the fibers obtained were investigated by scanning and transmission electron microscopy, as well as X-ray diffraction methods. It was shown that during fiber spinning stages the transformation of the TEOS drops into solid silica particles takes place as a result of hydrolytic polycondensation. This process can be characterized as the sol-gel synthesis. The formation of nano-sized (3–30 nm) silica particles proceeds without particles aggregation, but in a mode of the distribution gradient along the fiber cross-section leading to the accumulation of the silica particles either in the fiber center (wet spinning) or in the fiber periphery (mechanotropic spinning). The prepared composite fibers were carbonized and according to XRD analysis of carbon fibers, the clear peaks corresponding to SiC were observed. These findings indicate the useful role of TEOS as a precursor agent for both, silica in PAN fibers and silicon carbide in carbon fibers that has potential applications in some advanced materials with high thermal properties.

## 1. Introduction

Polyacrylonitrile (PAN) fibers are widely used in many sectors of the economy and industry due to the combination of high mechanical characteristics [1], wool-like properties [2], chemical resistance [3], good dyeability [4], and the possibility of using as precursors of high-strength carbon fiber [5]. First of all, such a variety of properties is due to the nature of acrylonitrile, which actively enters into polymerization reactions of various types (free-radical and anionic) in solutions, suspensions, and emulsions [3]. In addition, it perfectly interacts with various comonomers, in particular, with acrylates and carboxylates, which makes it possible to obtain various copolymers and fibers from them with the required structural and functional properties [6].

To change several specific properties of PAN fibers and films and give them special properties, the addition of various silicon-containing components has shown good efficiency. This approach makes it possible to impart hydrophobic properties to PAN-based materials while maintaining the inherent vapor- and air- permeability of PAN, which is required for textile, medical purposes, and the production of fabrics with special-purpose and self-cleaning materials [7,8,9]. In addition, modification of PAN with silicon-containing components is used in the production of membranes and filters to change their permeability and selectivity [10,11,12,13,14]. A surface modification of PAN fibers with silicon-containing coupling agents to improve compatibility with the matrix in the manufacture of the polymer composites was investigated in [15]. The particular interest presents the deposition of SiO_2_ coating on carbon fiber (CF) [16], and the modification of PAN fibers with silicon-containing additives for subsequent carbonization [17,18]. The combination of carbon structures and silicon carbide (SiC) in one fiber [11] leads to an improvement in the resistance to oxidation at elevated temperatures.

SiO_2_ is used as a silicon-containing modifier component, and composite fibers obtained by electrospinning [7,14,18,19,20,21], often, simultaneously with achieving the required special characteristics demonstrate inhomogeneous morphology due to the formation of large aggregates [7,8,12,14,20], that could decrease mechanical properties [12,22].

The possibility of obtaining monodisperse SiO_2_ particles with dimensions from 1 to 3 μm using the sol-gel method from tetraethoxysilane (TEOS) was shown in [23], etc. This method renders it possible to obtain homogeneous, well-dispersed SiO_2_ nanoparticles in a polymer matrix [24,25,26]. In existing papers, the production of hybrid PAN-SiO_2_ fibers, membranes, and films is carried out via the sol-gel method, using TEOS as a precursor of silica, obtaining either coating on the fiber/film [11,21], or via PAN synthesis in the medium of the sol formed from TEOS [27].

The main objective of this study was to investigate and demonstrate a novel method for the in-situ synthesis of particles into fibers during the spinning process with their cross-section distribution control. Until now, the area of obtaining composite PAN fibers from a mixed solution with TEOS remains unexplored. We have previously studied the rheological behavior and morphology of mixtures of TEOS with DMSO and solutions of PAN in DMSO [28]. The existence of regions of both mixed solutions and stable emulsions has been shown. The existence of the TEOS solubility region in PAN solutions in DMSO, as well as the existence of the stable emulsions with TEOS as disperse phase, opens the possibility of obtaining SiO_2_ in the fiber as a result of the hydrolytic polycondensation (HPC). This paper describes new methods for producing high-strength composite PAN-silica fibers from spinning solutions with TEOS, which are of interest for obtaining carbon-silicon carbide fibers with adjusted morphology.

## 2. Materials and Methods

### 2.1. Materials

PAN powder was purchased by Good Fellow Co., (Huntingdon, Great Britain) and contained 93.8% of acrylonitrile, 5.8% of methyl acrylate, and 0.3% of methyl sulfonate—with a molecular weight of 85 kg/mole and a polydispersity index of 2.1. DMSO (99.6% produced by EKOS-1, (Moscow, Russia) was dried using A4 molecular sieves (Sigma Aldrich, St. Louis, MO, USA) [29]. The residual moisture was determined by coulometric titration using the instrument Expert 007 M (produced by Econix Expert Ltd., Moscow, Russia). The residual moisture content in DMSO did not exceed 0.05%. TEOS (ECOS-1, Moscow, Russia) was used as an additive. A binary DMSO:distilled water solution (85:15, wt:wt) was prepared to simulate the precipitation process. 

### 2.2. Spinning Solution Preparation

PAN solutions with TEOS additions were prepared at 70 °C on a laboratory opposing mixer [22], which allows homogenizing concentrated polymer solutions, protecting them from ingress of air and moisture from the outside.

### 2.3. Fiber Spinning

The fibers spinning was performed on a multifunctional laboratory spinning line, which makes it possible to obtain fibers in various ways, such as wet, and mechanotropic.

#### 2.3.1. Wet Spinning

The spinneret with a diameter of 600 µm is immersed in a coagulation bath during the wet spinning process (DMSO:water baths compositions: 85:15; 80:20; 50:50, correspondingly) and the spinning solution supply unit is located horizontally (Figure 1). In the water washing bath, the fiber is washed from the solvent residues and goes to the drying rolls, the temperature of which is 80 °C, after that, it is thermally stretched at 100 °C in an oven and goes to the thread spreader. Usually, the speed at each stage is higher than at the previous one. The feed rate of the solution was 0.012 m/min, the temperatures of the drying drums and thermal stretching, as well as the speed of rotation of the winding at each stage, were adjusted depending on the properties of the spinning system and fiber and taking into account the requirements for the degree of drawing.

#### 2.3.2. Mechanotropic Spinning

The mechanotropic method was developed earlier in our laboratory [30], and favorably differs from the wet and dry-wet jet methods by the absence of coagulation precipitation baths, which makes the spinning process more environmentally friendly and economically profitable, since no need to recover large volumes of coagulation baths for extracting the useful for technology and harmful for ecology PAN solvents. The phase separation occurs in the polymer solution when the jet is stretched [31], the polymer chains draw and approach the center of the jet, with the formation of a solid continuous fiber, and the solvent phase migrates on its surface. 

During mechanotropic spinning (see Figure 2), the spinning dope feed unit is positioned vertically providing a solution linear rate of 0.075 m/min from a spinneret hole with a diameter of 500 µm.

### 2.4. Film Preparation

A 200 μm thick film was prepared for studying morphology by pouring the solution onto a glass surface and drying it gradually by increasing the temperature from 30 to 80 °C until a constant weight was obtained. The heating rate was 10 °C over 3 h. The PAN concentration in the solution was 20%, and TEOS was 5% relative to DMSO. To proceed with the TEOS hydrolysis-condensation reaction, the film was kept in boiling water for 12 h and then dried at 100 °C until a constant weight was achieved.

### 2.5. Mechanical Characteristics of Fibers

The mechanical properties of the fibers were studied using an Instron 1122 tensile testing machine on the 10 mm monofilament length with a stretching rate of 10 mm/min. All measurements were conducted at room temperature. The fiber diameter was determined using the aforementioned optical microscope. To obtain the mechanical characteristics for each fiber batch, data were averaged over at least 10 filaments.

### 2.6. Synchronous Thermal Analysis

DSC and TGA measurements were performed using a TGA/DSC 3+ synchronous thermogravimetric analyzer (Mettler Toledo, Greifensee, Switzerland). The samples were heated from 25 to 1000 °C at a rate of 10 K/min in a nitrogen atmosphere with a flow rate of 70 mL/min. The measurement results were analyzed using the STARe service program provided with the instrument. The weighting accuracy was 0.0025%, while the accuracy of temperature and enthalpy measurements was ±0.3 °C and ±1 J/g, respectively.

### 2.7. Microscopy

Before studying fibers containing TEOS by electron microscopy, they were first subjected to 4 h of boiling water treatment to enable the hydrolytic polycondensation reaction of TEOS and form silicon dioxide.

#### 2.7.1. Transmission Electron Microscopy (TEM)

The study was performed using a transmission electron microscope LEO 912 ab omega (LEO Carl Zeiss SMT Ltd., Jena, Germany) operating at an acceleration voltage of 100 keV. For analysis, fiber, and film samples were prepared as thin sections (100 nm) placed on a copper mesh substrate. To observe the morphology of fiber cross-sections, the bundle of fibers was first mixed with an epoxy binder to obtain a microplastic cylinder with a diameter of ~500 μm. Then a cut was made perpendicular to the long axis of the fibers bundle using an Ultracut-R ultramicrotome (Leica Microsystems, Wetzlar, Germany).

In this study, we utilized electron energy loss spectroscopy (EELS) to analyze the composition of our samples. EELS is a powerful technique that allows for the determination of the energy loss of electrons as they pass through a material, providing information on the material’s composition and electronic properties. In this method, a high-energy electron beam is focused onto a thin sample, and the scattered electrons are analyzed using an electron spectrometer. The technique is highly sensitive to changes in electronic structure, making it ideal for the characterization of materials with complex electronic structures. The EELS measurements were carried out at a beam energy of 100 keV.

#### 2.7.2. Scanning Electron Microscopy (SEM)

The fiber morphology was analyzed using a Versa 3D DualBeam electron-ion microscope (FEI, Hillsboro, OR, USA) using a low vacuum secondary electron detector (LVSED) at 100 Pa, and equipped with an energy-dispersive X-ray spectroscopy (EDS) detector, operating at accelerating voltages of 5 and 10 kV. Elemental analysis data obtained from electron microscopy images were processed using the EDAX TEAM software. The samples were fixed vertically onto carbon tape. To obtain a transverse cleavage, a bundle of fibers was cryo-fractured to obtain brittle cleavages (Figures 12 and 14a), or the fibers were cut by ultramicrotome EM UC7 (Leica Microsystems, Wetzlar, Germany) (Figure 14b).

### 2.8. Coagulation

To assess the influence of TEOS on the coagulation process the modeling of wet spinning was performed on a droplet of the solutions placed in a gap of constant thickness between a slide and cover slip and surrounded with a coagulant. The coagulant used was a mixture of DMSO and water with 85:15 ratio of components. The detailed methodology is described in [32].

### 2.9. Rheological Measurements

The rheological properties of the solutions were tested using a rotational rheometer, HAAKE MARS 60 (Thermo Fisher Scientific, Karlsruhe, Germany), in continuous and oscillatory modes of shear deformation with a cone-plate geometry (cone angle 1°). A 20 mm cone was used for 30% and 33% PAN solutions, while a 60 mm cone was used for 20% and 27% PAN solutions. Flow curves were obtained by a stepwise increase of shear rate in the range of 10^−3^–10^3^ s^−1^ until the sample began to flow out of the gap. The frequency dependences of the storage and loss moduli were measured in the linear viscoelasticity region at frequencies ranging from 0.628 to 628 rad/s. All measurements were performed at 70 °C.

### 2.10. Optical Microscopy

The fiber diameter was measured using a Biomed 6PO microscope (Biomed Co., Moscow, Russia) equipped with a ToupTek E3ISPM5000 camera (ToupTek Photonics Co., Hangzhou, China).

## 3. Results and Discussion

### 3.1. PAN-DMSO-TEOS Spinning Solution Rheology

Preparing PAN solution with a soluble silicon-containing compound in a common solvent allows for the formation of a homogeneous mixture, where components are distributed on a molecular level. TEOS was selected as a model additive due to its limited compatibility with PAN solutions in DMSO [28] and its ability to form nanometer-sized SiO_2_ particles during fiber spinning and the HPC reaction [33,34].

Based on previous results [28], systems with TEOS content of 5% on DMSO, which falls within the solubility region, and 15% that lies in the region of a stable emulsion, were chosen to obtain fibers by mechanotropic and wet spinning methods. Table 1 shows the compositions of spinning systems, including the TEOS content as a percentage of DMSO content, and its corresponding content in the finished fiber.

Mixtures with 33% PAN at room temperature quickly form a gel [35] due to nitrile-nitrile interactions in a concentrated solution. Therefore, to prevent gel formation during rheological measurements and fiber spinning procedures, the temperature inside the dosing device was maintained at 70 °C. Figure 3 illustrates the flow curves and viscoelastic properties of the spinning solution at 70 °C.

The influence of TEOS on rheological characteristics diminishes as polymer concentration increases. The addition of TEOS to a 20% solution results in a 26% decrease in viscosity, whereas for 30% and 33% solutions, the reduction in viscosity is no more than 3%. The same trend is observed for the frequency dependencies of complex elastic modulus components. At a PAN concentration of 33%, the effect of TEOS is minimal, leading only to a slight change in the slopes for loss and storage moduli. These findings indicate that highly concentrated spinning systems with TEOS exhibit behavior similar to solutions without additives since the presence of TEOS does not significantly affect their rheological characteristics.

### 3.2. PAN-TEOS Film Morphology

Figure 4 illustrates the morphology of a film that was cast from a solution containing 20% PAN-DMSO and 5% TEOS (on DMSO).

The film contains inclusions of round shape uniformly distributed throughout the sample, with a size range from 20 to 60 nm. This means that silicon-containing particles can be immersed in PAN by using the common solvent both for polymer and TEOS. During the film’s drying process, while the solution is still in a liquid state, the mobility of TEOS and its hydrolysis-condensation products is relatively high. Initially, TEOS dissolves uniformly but then undergoes sole-gel transformation and forms oligosiloxanes which are isolated into a separate phase. They then merge into droplets with size increasing up to the deep stages of the hydrolysis-condensation reaction. At this point, poly(oligosiloxanes) are forming and eventually converting into solid SiO_2_ inclusions. A similar process is described in [25,26], where silicon-containing particles were obtained in an epoxy matrix.

One of the useful methods that could indicate the chemical composition of obtained particles in the PAN films is electron energy loss spectroscopy (EELS) to collect an energy loss spectrum. EELS is a technique used to analyze how initially almost monoenergetic electrons’ energy spreads after interacting with a sample. When fast electrons pass through the matter, they experience numerous scattering events with some transmitted electrons not suffering any energy loss (known as elastic scattering). Elastic scattering contributes the most input to contrast in conventional transmission electron microscopy (TEM) images. However, in EELS, the technique focuses on analyzing the energy spectrum of inelastically scattered electrons, which have undergone energy loss processes. Thus, EELS provides a way to study a sample’s electronic structure and composition by examining how it affects the energy of the scattered electrons.

Silica exists in tetrahedron form which may join together to form a large number of crystalline silica polymorphs [36], as well as amorphous and glassy phases. The SiO_2_-contained minerals were studied by EELS methods [37]. It was found the Si L23 edges from quartz and the clay minerals exhibit very similar edge shapes, characterized by two initial sharp peaks: A (106.3) and B (108.4 eV), followed by a sharp peak C (115.0) and a broad peak D (ca. 133 eV). The observed structures have been assigned to the unoccupied molecular orbitals of a SiOX- tetrahedron [38].

Figure 5 shows the EELS spectrum of a PAN film annealed at 150 °C for 10 h for TEOS hydrolysis compared with data for Si L23 in SiO_2_ transitions obtained by Gatan company [39].

In the figure, are presented the EELS spectra in the 100–150 eV range, dominated by the L2 edge transition of Si at 110 eV. Also here clearly seen a mild peak at 115 eV and a broad peak at 130 eV, which quite well corresponds with the comparison data and the data presented in [37] for quartz and some SiO_2_-contained minerals. From the analysis of the spectra comparison and the reference data, it can be seen that obtained particles consist of SiO_2_.

Based on the experiment described above, a conclusion can be drawn regarding the formation of inclusions of a similar nature in the resulting fiber. But the size of the particles could be different because the formation of a gel fiber during spinning occurs much faster than the process of obtaining a film, which limits the possibility of coalescence of the dispersed TEOS and its hydrolysis/condensation products, and the subsequent formation of a solid fiber. Therefore, additional experiments are required to investigate the morphology of the fiber and confirm the formation of similar inclusions.

### 3.3. PAN-TEOS Fiber Spinning

To investigate the impact of the TEOS additive on the fiber formation process, a series of fibers were spun from the prepared solutions via mechanotropic and wet spinning, and their properties were analyzed. Analysis of the phase separation mechanisms during different methods of fiber spinning led to the assumption that depending on prevailed directions of the diffusion streams the attainment of a diverse distribution (as shown in Figure 6) of the inclusions throughout the fiber’s cross-section is possible.

To validate this hypothesis, the test spinnings were conducted, and the properties of the fibers were analyzed.

#### 3.3.1. Fibers with the Surface Layer Enriched with SiO_2_ Particles (Mechanotropic Spinning)

Fibers were produced from a spinning solution of PAN-DMSO-TEOS with a polymer content of 33% and TEOS concentrations of 5% and 15% (relative to DMSO)—hereafter referred to as 10% and 30% in terms of polymer, respectively. According to [28], the first composition is a solution, while the second one is an emulsion. The spinning conditions are outlined in Table 2. To assess the impact of the additive on fiber properties, a sample without TEOS was produced using dope with the same polymer content (33%) and under similar spinning conditions. The presence of TEOS did not affect the fiber spinning process in the case of complete solubility, but somewhat reduces the limiting draw ratio when spinning from an emulsion.

The bundle of spun fibers was analyzed by optical microscopy under transmitted light, and the corresponding images are presented in Figure 7.

The fibers produced from PAN neat solution and solution initially containing 10% of TEOS per polymer (i.e., 5% per DMSO) are transparent, without visible defects, and had a smooth surface with uniform thickness. In contrast, fibers produced from an emulsion with initial TEOS content of 30% per polymer (i.e., 15% per DMSO) exhibited weak light scattering due to the presence of the second-phase particles. The inhomogeneity caused light scattering and “turbidity”, resulting in an opaque fiber. 

The mechanical properties of the fibers are presented in Table 3. 

The fibers obtained exhibit excellent mechanical properties, with a strength of 800 MPa for the fiber without TEOS, and only a slight reduction to 700 MPa with the introduction of the additive. 

Images of thin transverse sections of the fibers were examined using TEM at various magnifications, and these results are presented in Figure 8.

The obtained fibers have regular round cross-sections without any defects. Numerous SiO_2_ like particles ranging in size from 3 to 30 nm are seen in the fibers, formed from TEOS due to the HPC reaction occurring within the spun fiber. The distribution of particles is uneven across the fiber cross-section. Only a few particles are observed in the core of the fiber, while at the periphery, particles are distributed at a distance of 5–50 nm from each other without aggregation.

This distribution can be explained by the mechanotropic process of fiber formation: during the phase separation of the polymer solution, the DMSO solvent moves from the center to the surface of the fiber and pulls TEOS drops, leading to a decrease in concentration at the center and an increase at the periphery. This presents an opportunity to obtain fibers with a gradient distribution of the silicon-containing component and an increased concentration of particles near the fiber surface.

#### 3.3.2. Fibers with Core Enriched with SiO_2_ (Wet Spinning)

In the case of the wet spinning method, the mechanism of the solution coagulation is determined by the kinetics of interdiffusion of the coagulant into the fiber and the solvent outward, which differs significantly from the mechanotropic method, where a predominantly unidirectional process of solvent release to the outside takes place.

To obtain the most homogeneous defect-free fiber, the softness of the coagulant for the PAN solution was selected using the example of a 20% of PAN solution in DMSO. The choice of a lower polymer concentration in the solution is due to the possibility of stable droplet formation in a narrow gap of the measuring cell [32] and to the exclusion of gelling the fiber surface with air moisture to obtain comparable and well-reproducible results. 

The model drop method makes it possible to study the effect of changing the composition of the coagulant on the resulting morphology of the fiber cross-section model. Figure 9 shows the result of the interaction of 20% of PAN-DMSO solution with some coagulants prepared with a mixture of water with DMSO in different ratios.

Water is a highly rigid coagulant that causes the rapid formation of numerous large voids or vacuoles. However, the introduction of DMSO results in a proportional reduction in the number and size of these defects. The optimal composition for achieving soft, defect-free coagulation without dissolution of the droplet is a coagulant consisting of DMSO and water in a ratio of 85:15. This precise composition was used as the coagulant in the first bath for the wet method of fiber spinning. 

TEOS is a hydrophobic liquid that is not compatible with water, and this feature determines the kinetics of its diffusion upon interaction with a water-containing coagulant. In the case of TEOS in a DMSO: water mixture, it is initially soluble in DMSO, but as the front of water penetrates the droplet, TEOS transforms into insoluble microdroplets as the disperse phase of emulsion which move together with water towards the center of the fiber, while the solvent moves in the opposite direction. This means that a front of coagulant diffusion in the solution jet is prevailing, i.e., leads to producing fibers with a high content of silicon-containing components in the center. We have previously simulated this phenomenon of TEOS redistribution during coagulation using the method of optical interferometry [28]. Figure 10 shows the result of the experiment confirming the accumulation of TEOS microdroplets in the middle of the solution drop. The second phase droplets have a regular, round shape, with a size of up to 2 μm.

Using the results of preliminary experiments, we prepared a dope with a PAN content in DMSO of 27% containing 5% TEOS relative to DMSO, which corresponds to 13.5% relative to the polymer. The spinning conditions are provided in Table 4.

Figure 11 shows images of the fibers obtained by wet spinning.

The use of a soft coagulant and appropriate spinning conditions results in fibers that are uniform in thickness and almost transparent. However, the wet process can lead to a slight “turbidity” in the fibers, which can be classified as translucent, possibly due to the formation of a denser shell, and a more defective surface compared to the mechanotropic spinning, as was demonstrated above. Table 5 presents the mechanical properties of the fibers.

A TEOS addition does not change the mechanical characteristics of the fibers; their strength is 500 MPa, and the elastic modulus is 5.5 GPa.

### 3.4. Composite Fibers Morphology Comparison

To compare containing TEOS fibers obtained by mechanotropic and wet methods, we determined the silicon content in different areas of the fiber cross-section using an SEM equipped with an EDS detector. Due to the significant accumulation of charge in the fibers under examination, a complete mapping of the silicon distribution was not feasible. As a result, the determination of the silicon content in the fibers was carried out in a mode “region by region”. We chose areas starting from the central part of the cleavage and moving toward the surface, to observe changes in the silicon concentration across the cross-section. These areas are indicated in Figure 12.

The data obtained confirm a gradient distribution of the silicon-containing component across the cross-section of the fibers produced by mechanotropic and wet spinning methods. In the case of the mechanotropic method, the core of the fiber contained a maximum of 3.6% silicon by weight; the region between the core and surface contained 4.4–5.6%, while the surface contained 6.7%. For the wet spinning method, near the fiber surface, it contains 1.3–1.8% of silicon by weight, while the core contained 0.8–1%. 

The Si wt.% values obtained from both methods were plotted along the fiber radius on the same graph in Figure 13.

The relationship between the content of silicon along the fiber cross-section (as shown in Figure 13) indicates that the method of fiber production, and therefore, the mechanism of phase separation in the solution jet, predetermines the gradient distribution of SiO_2_ particles in the fiber. In wet spinning, the fiber core is enriched with silicon, while in mechanotropic spinning, the surface is enriched. In both methods, a twofold difference in silicon concentration between the center and periphery was observed along the fiber cross-section.

This morphology is not only interesting from the perspective of the properties of the white fiber but also for further carbonization to produce a hybrid carbon-silicon carbide fiber that will contain particles either in the near-surface layers or in the core of the fiber.

Figure 14 shows the fiber cross-section and surface morphology of PAN-TEOS fibers obtained by mechanotropic and wet methods.

The spinning method is expected to affect the surface morphology of the fibers. For the wet method, the fibers have a more uneven surface (Figure 14a), and numerous caverns and sagging along the fiber axis are visible, which is due to the interaction of the polymer solution jet with the coagulant. In contrast, the fiber obtained by the mechanotropic method has a smooth and even surface (Figure 14b).

### 3.5. Thermal Behavior of Fibers

The composite fibers were heat-treated in Ar using a synchronous thermal analysis device. Figure 15 shows the data on weight loss (a) and thermal effect in the PAN cyclization region (b).

The TGA/DSC data indicate that there are no residual traces of solvent or water in the samples, and the introduced modifier components do not affect the cyclization of PAN fibers. Up to 600 °C, the additives do not affect the weight loss of the samples. It is noteworthy that there are no thermal effects or mass changes when comparing the sample without additives to the one with TEOS, indicating that the HPA reaction in the studied fiber was completed by the time of STA.

The PAN film with TEOS and the spun precursor fiber were carbonized by heating them to 1600 °C in Ar. The resulting samples were analyzed by X-ray diffraction method (Figure 16).

The diffraction patterns demonstrate clear reflections of the crystal lattice of silicon carbide—Moissanite, particularly at angles 2θ = 36°. Although the X-ray pattern contains significant noise due to the small amount of the additive, the presence of peaks intrinsic for silicon carbide at angles 2θ 36°, 60°, and 72° is visible.

## 4. Conclusions

The introduction of TEOS in PAN allows for the production of fibers with a predetermined gradient distribution of individual SiO_2_ like nanoparticles formed from TEOS: with increased particle content at the surface in the case of mechanotropic spinning, or in the core in the case of wet spinning.

Using the mechanotropic method the proportion of Si in the fiber increases from the center to the edge from 3.6 to 6.7 wt%, respectively. Such fibers have high tensile strengths of ~700 MPa and smooth, defect-free surfaces, similar to fibers obtained under the same conditions without additives.

Using the wet spinning method, a high content of the silicon-containing phase is accumulated in the center of the fiber. With an initial average content of TEOS 13.5% per polymer, the proportion of silicon increases from 0.8 to 1.8 wt% from the edge to the center. The presence of TEOS in PAN fibers does not affect the mechanical properties. The resulting fibers have a tensile strength of 500 MPa.

Thermolysis of composite samples by heating to 1600 °C leads to the formation of silicon carbide in the carbon fibers.

## Figures and Tables

**Figure 1 polymers-15-02579-f001:**
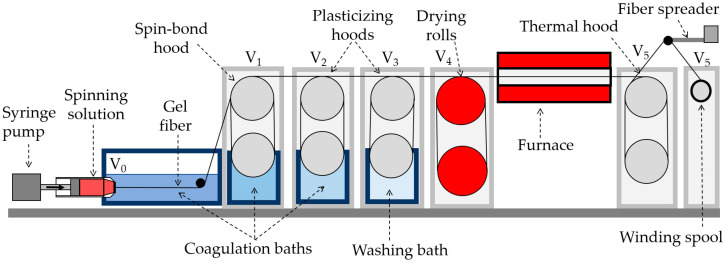
Scheme of a laboratory line for the wet spinning.

**Figure 2 polymers-15-02579-f002:**
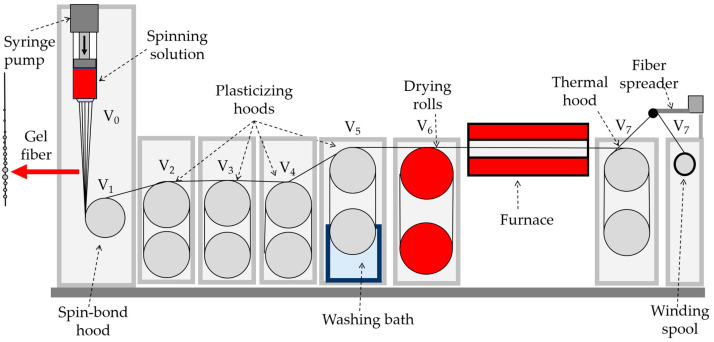
Scheme of a laboratory line for mechanotropic spinning.

**Figure 3 polymers-15-02579-f003:**
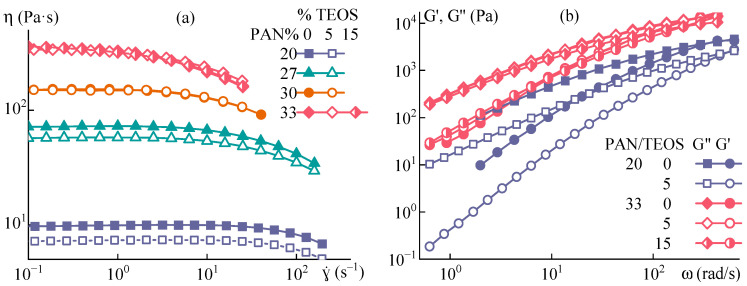
(**a**) Flow curves and (**b**) frequency dependences of dynamic moduli for the spinning PAN-DMSO solutions with TEOS at 70 °C. PAN and TEOS concentrations are shown in the figures.

**Figure 4 polymers-15-02579-f004:**
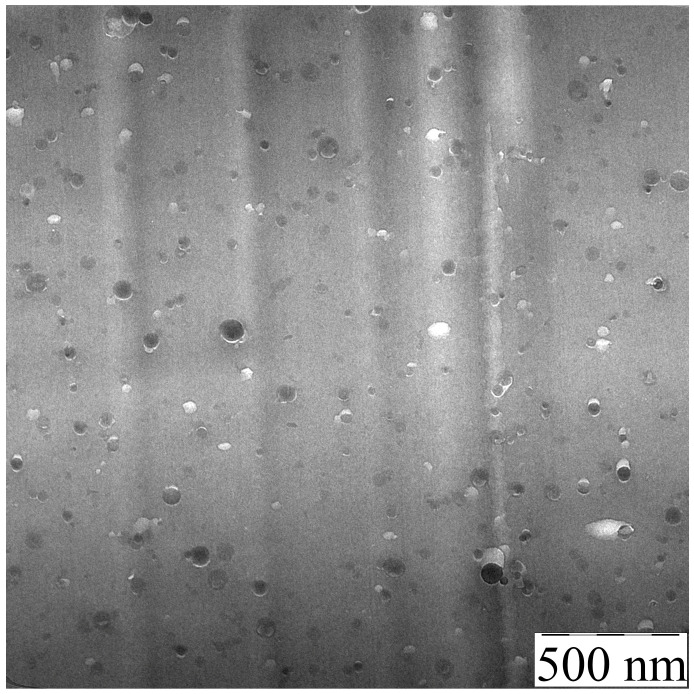
TEM image of a film section that was cast from a PAN solution containing 5% TEOS.

**Figure 5 polymers-15-02579-f005:**
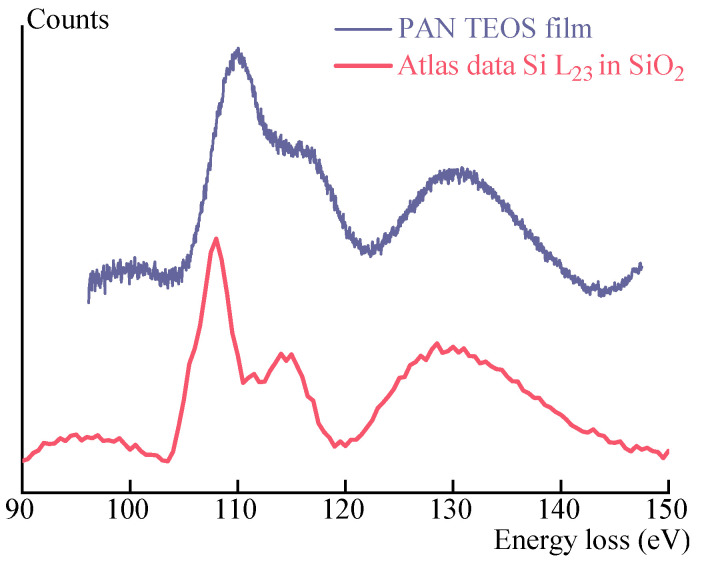
Si L23 edges of the film obtained from a PAN-DMSO solution containing 5% TEOS and comparison spectra from a Si L23 region in SiO_2_.

**Figure 6 polymers-15-02579-f006:**
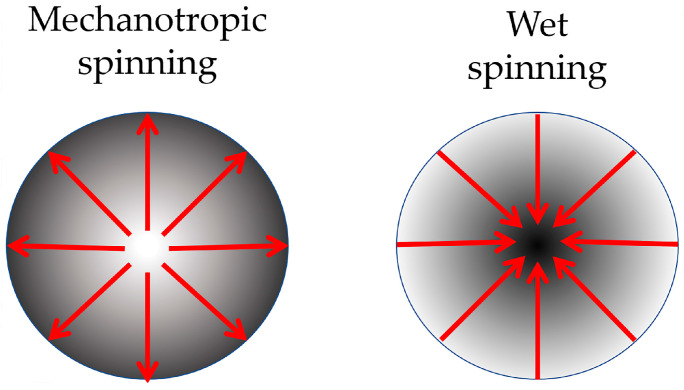
Fiber cross-section model. Red arrows show the movement of the TEOS over the fiber cross-section during mechanotropic and wet spinning processes. The dark color indicates the accumulation of silicon-containing particles in the polymer matrix.

**Figure 7 polymers-15-02579-f007:**
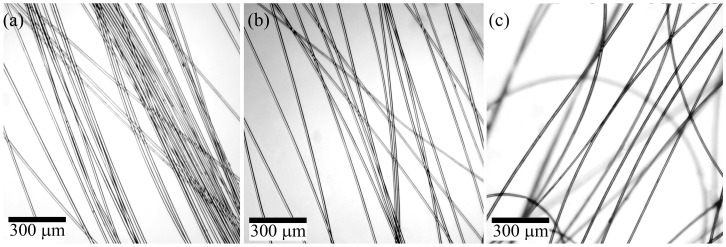
Images of fibers, contained: (**a**) 0%; (**b**) 10%; (**c**) 30% of added TEOS per PAN. Images were obtained by transmitted optical microscopy.

**Figure 8 polymers-15-02579-f008:**
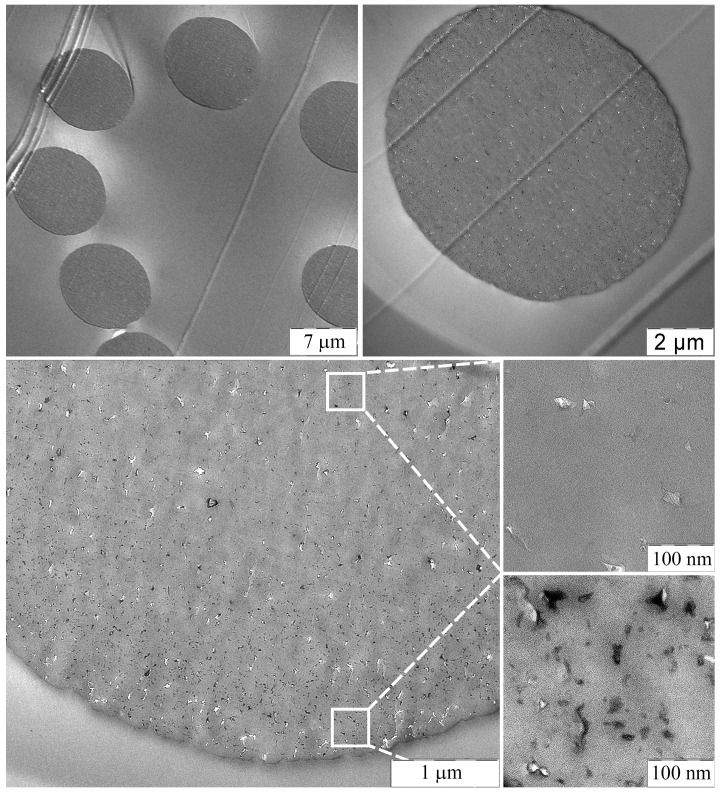
Cross-section images of a bundle of fibers containing 10% of TEOS (per polymer) obtained by TEM at various magnifications.

**Figure 9 polymers-15-02579-f009:**
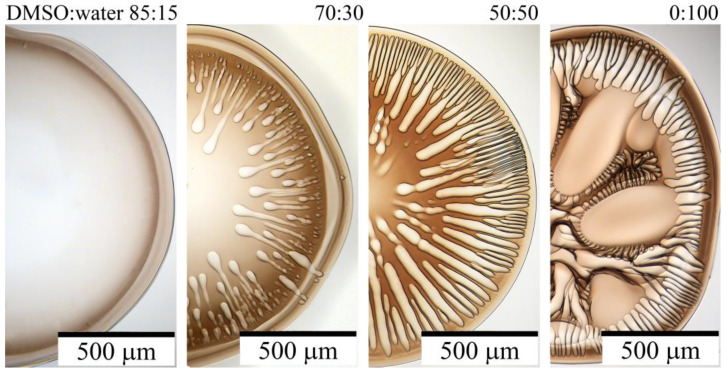
The final stages of the droplet of 20% PAN-DMSO solution coagulation. Water and DMSO concentrations in the coagulant are shown in the figure obtained by transmitted optical microscopy.

**Figure 10 polymers-15-02579-f010:**
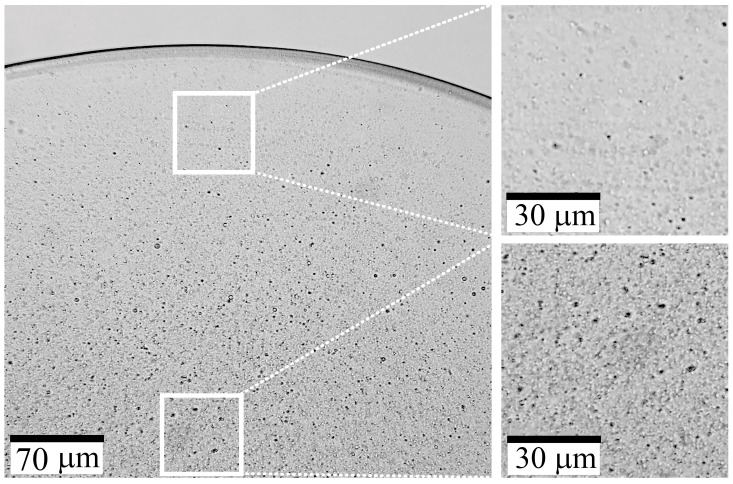
Coagulated PAN droplet containing 10% TEOS with enlarged areas near the edge and in the center. Images were obtained by transmitted optical microscopy.

**Figure 11 polymers-15-02579-f011:**
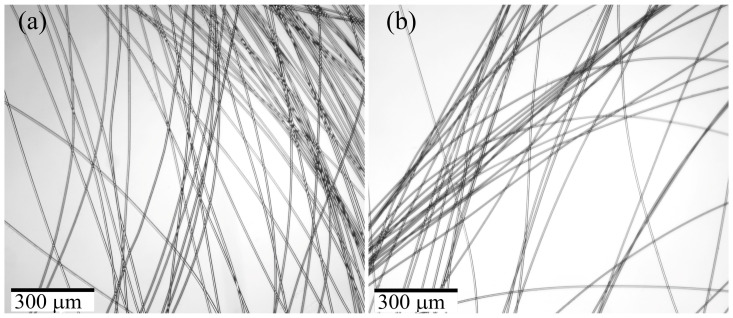
PAN fibers obtained by the wet method (**a**) without TEOS and (**b**) containing 13.5% TEOS. Images were obtained by transmitted optical microscopy.

**Figure 12 polymers-15-02579-f012:**
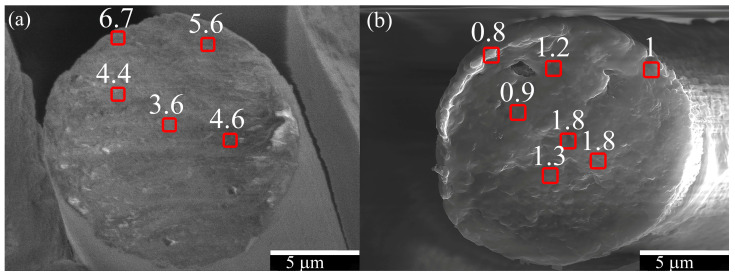
Images of fiber cleavages (**a**) PAN + 30% TEOS mechanotropic spinning; (**b**) PAN + 13.5% TEOS wet spinning. Areas in which the silicon content was determined are marked in red, the numbers indicate the Si wt.%. Images were obtained by scanning electron microscopy.

**Figure 13 polymers-15-02579-f013:**
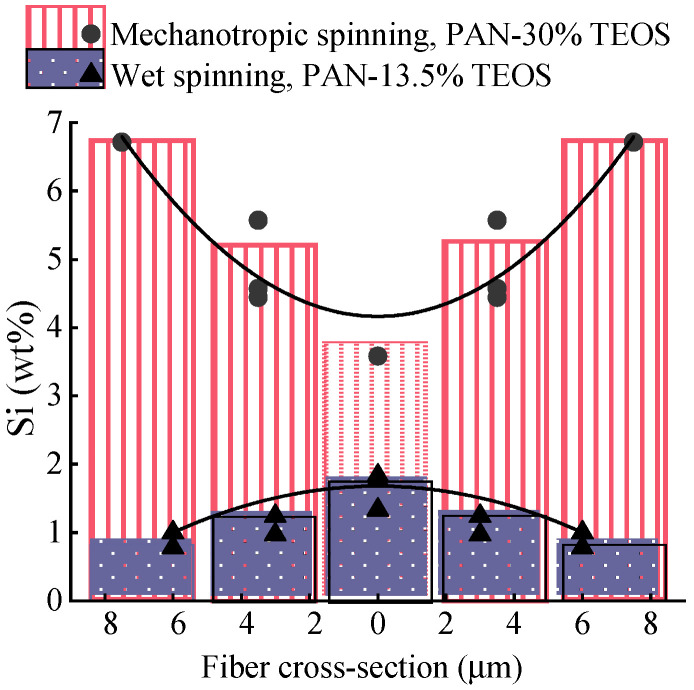
Dependences of silicon content (wt.%) on cross-section radius for fibers produced using mechanotropic and wet spinning methods.

**Figure 14 polymers-15-02579-f014:**
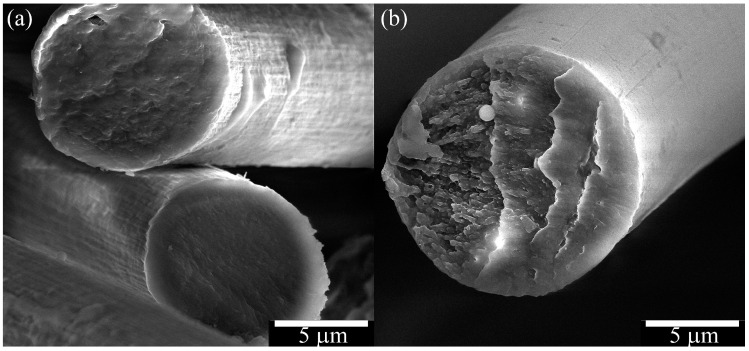
Morphology of the surface and cross-sections of fibers obtained by (**a**) wet PAN with 13.5% TEOS, (**b**) mechanotropic PAN with 30% TEOS. Images were obtained by scanning electron microscopy.

**Figure 15 polymers-15-02579-f015:**
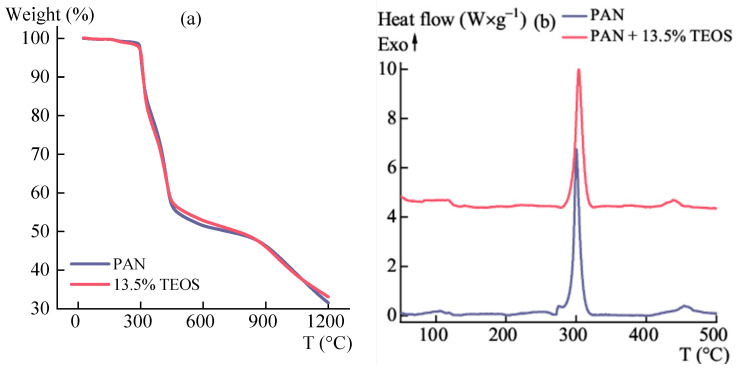
Temperature dependences of (**a**) mass loss and (**b**) heat flux for PAN fibers containing TEOS. The black arrow shows the heat flow direction.

**Figure 16 polymers-15-02579-f016:**
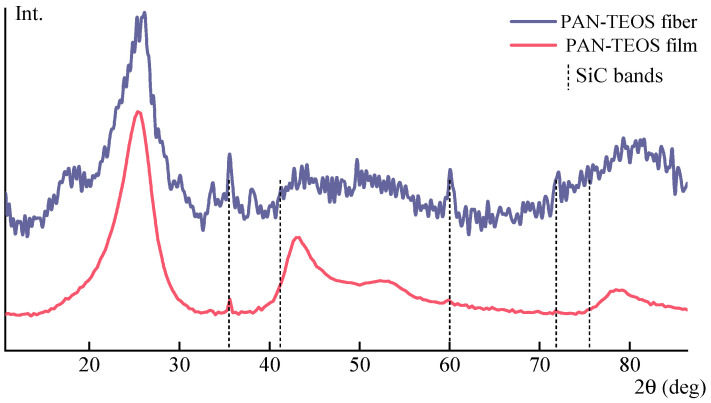
X-ray patterns of carbonized samples after heat treatment up to 1600 °C of a composite film and PAN fiber with initial concentrations of TEOS of 13.5 per film and 30 per fiber.

**Table 1 polymers-15-02579-t001:** Composition of spinning solution and spun fiber, where 0 is a neat solution.

Spinning Solution			Fiber
PAN%	DMSO%	TEOS (% to DMSO)	TEOS (% to PAN)
27	73	0	0
		5	13.5
30	70	0	0
		5	11.6
33	67	0	0
		5	10
		15	30

**Table 2 polymers-15-02579-t002:** Mechanotropic spinning parameters.

Sample	Linear Winding Speed, m/min							Draw Ratio	
	V1	V2	V3	V4	V5	V6	V7	Relative V1	Full
PAN	4.3	4.6	9	10.5	11.6	14.1	26	6 ^1^	330 ^2^
PAN-TEOS 10%	4.2	4.7	10.2	10.9	11.6	14.1	26	6.2 ^1^	330 ^2^
PAN-TEOS 30%	0.9	2.2	3.2	7.1	7.8	8.1	10	11 ^1^	200 ^3^

^1^ V7/V1. ^2^ V7/0.08. ^3^ V7/0.05.

**Table 3 polymers-15-02579-t003:** Mechanical characteristics of PAN fibers with TEOS obtained by the mechanotropic method.

Mechanotropic Spinning	PAN	PAN-TEOS (10%)	PAN-TEOS (30%)
Fiber diameter, μm	13 ± 3	7 ± 2	13 ± 1
Tensile strength, MPa	800 ± 90	700 ± 70	700 ± 80
Elongation at break, %	18 ± 3	19 ± 4	16 ± 2
Elastic modulus, GPa	6 ± 1	8 ± 2	13 ± 3

**Table 4 polymers-15-02579-t004:** Wet spinning parameters.

Sample	Linear Winding Speed, m/min					Draw Ratio	
	V1	V2	V3	V4	V5	Relative V1	Full
PAN	2.1	4.8	5.8	6.1	9.5	4.5 ^1^	80 ^2^
PAN-TEOS 13.5%	2.1	4.8	5.8	6.1	9.5	4.5 ^1^	80 ^2^

^1^ V5/V1. ^2^ V5/0.12.

**Table 5 polymers-15-02579-t005:** Mechanical characteristics of PAN fibers with TEOS obtained by the wet method.

Wet Spinning	PAN	PAN-TEOS (13.5%)
Fiber diameter d, μm	12 ± 3	11 ± 0.3
Tensile strength, MPa	500 ± 30	500 ± 20
Elongation at break, %	20 ± 1	19 ± 2
Elastic modulus, GPa	5.5 ± 0.5	5.5 ± 0.5

## Data Availability

The data that support the findings of this study are available from the corresponding author upon reasonable request.

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
