# Peer review of "Polyacrylonitrile Fibers with a Gradient Silica Distribution as Precursors of Carbon-Silicon-Carbide Fibers"

_polymers, 2023, doi:10.3390/polym15112579_

Round 1

Reviewer 1 Report

The work on the title Polyacrylonitrile Fibers with a Gradient Silica Distribution as Precursors of Carbon-Silicon-Carbide Fibers” is a good piece of research work. The authors have prepared  polyacrylonitrile (PAN) fibers containing tetraethoxysilane (TEOS)  by two methods namely wet spinning method and mechanotropic spinning method. In both methods the authors found the gradient distribution of silica in the spun fibre and the graduation of silica distribution in fibres by two methods in opposite direction that is also interesting. The manuscript can be accepted for publication. However, I would like to suggest to do some corrections before accepting. My comments and suggestion are as follows:

Line 51-52: The authors wrote a sentence, A surface modification of PAN fibers with silicon

containing.............was investigated in [15]. The authors should mention the name of investigators.

And it should be mentioned other similar sentences.

Line 99-106: The authors should correct Grammatical error.

Line 105: Feed rate should be corrected.

Line 309-311: The authors should clearly mention the concentration of solution using a standard method of expression.

Table 3: In mechnotropic method, inclusion of TEOS in PAN increased its elastic modulus and it is a general trend. However, in wet method, inclusion of TEOS did not improve the elastic modulus. why?

Moreover, the values of tensile strength are reported with a very high value of standard deviation. It seems the fibre is not homogeneous and the strength is not same throughput the length of fibre.

Line: 319-320: Reduction of strength  100 MPa,  is not really slight?

Figure: What kind of images in the figure?

Line: The authors should write everywhere either DMSO: H2O or H2O: DMSO but not randomly.

Figure 14, Line 435: How was the cross section of fibres obtained? How was broken?

Axis caption of the figures should write properly with their units

English is good

Author Response

Dear colleague,

Thank you for taking the time to review our publication and for your thorough analysis. We have carefully considered all of your questions and provided detailed responses below.

Question:

However, I would like to suggest to do some corrections before accepting.

My comments and suggestion are as follows:

Line 51-52: The authors wrote a sentence, A surface modification of PAN fibers with silicon containing.............was investigated in [15]. The authors should mention the name of investigators.

And it should be mentioned other similar sentences.

Answer:

With all due respect to the reviewer, it is important to adhere to the MDPI rules, which state that reference numbers should be placed in square brackets [ ] within the text. While it is understood that many investigators contributed to the referenced paper, it is not feasible to mention all of them individually. However, solely mentioning the first author may be considered disrespectful to the other investigators involved.

 Question:

Line 99-106: The authors should correct Grammatical error.

Answer:

Could you indicate that error? We checked the text and didn't find it.

 Question:

Line 105: Feed rate should be corrected.

Answer:

The feed rate was fixed.

Question:

Line 309-311: The authors should clearly mention the concentration of solution using a standard method of expression.

Answer:

The caption was modified.

Question:

Table 3: In mechanotropic method, inclusion of TEOS in PAN increased its elastic modulus and it is a general trend. However, in wet method, inclusion of TEOS did not improve the elastic modulus. why?

Moreover, the values of tensile strength are reported with a very high value of standard deviation. It seems the fibre is not homogeneous and the strength is not same throughput the length of fibre.

Answer:

This is a really interesting result that requires more careful verification. It is known that the addition of solid particles into the polymer matrix increases the modulus of elasticity.

The difference for two spinning modes can be explained by two factors:

1) This is probably due to the distribution of particles over the cross-section - in the mechanotropic method, they are concentrated near the surface, and in wet spinning mostly in the volume of the fiber. The data obtained can be indicator of more sensitive reinforcing effect in the case of accumulation of filler near the fiber surface. 

2) In addition, during fibers processing the liquid droplets of TEOS are undergone hydrolytic condensation and their transformation to the solid SiO2 particles.  At the surface, the likelihood of contact with air moisture is higher - and completeness of such transition is more probable.

Indeed, we observe a large scatter in the values, which is due to several factors: work with small volumes of solutions, only partial degassing and filtration of the dopes. These factors could lead to the local inhomogeneities in the fiber, which affect the reproducibility of results. Nevertheless, we always take a wide number of samples from different sections of the thread, to obtain the most reproducible result, and that is why the spread is quite large. But in one section of the fiber the properties are more uniform.

Question:

 Line: 319-320: Reduction of strength  100 MPa,  is not really slight?

Answer:

We think that attention should be paid to absolute strength values reducing from 800 to 700 MPa which is about 14% compared with the neat fiber. It is not so much because common PAN fibers as usual have a strength of about 400 - 700 MPa.

Question:

Figure: What kind of images in the figure?

Answer:

We didn't understand the question. In which figure? Judging by the sequence of questions, we are talking about one of the figures 9-12. Survey methods have been added to the captions for these drawings.

 Question:

Line: The authors should write everywhere either DMSO: H2O or H2O: DMSO but not randomly.

Answer:

We revised the Manuscript. Now it is written DMSO:water through all text.

Question:

Figure 14, Line 435: How was the cross section of fibres obtained? How was broken?

Answer:

Fiber cross-sections were made using an ultramicrotome, that information now is given in the experimental part above.

Question:

Axis caption of the figures should write properly with their units

Answer:

We revised the Manuscript and corrected all captions.

Reviewer 2 Report

It is interesting that the PAN fibers containing Si given by two routes were compared completely. However, some issues need to be cleared or revised.

1. The introduction of silicon source cannot result in obvious or special functionality of PAN fibers in the elucidation of Introduction part. And factually, this research showed this point that the strength of PAN fiber was not improved. Why do the authors introduce Silicon into PAN fibers? Or you should explain or evaluate further likely “novel” functions of as-spun PAN fibers you made.

2. Based on the spinning processes, the decomposition of TEOS to form SiO2 is not easy. It’s well-known that the model process of casting film is very different from fiber forming process. The TEOS would be evaporated easily in the spinning process? The silicon of inner fibers identified by EDS is hard to be attributed to SiO2.

3. It is better if stress-strain curves were attached for Mechanical characteristics except for Table 5.

4. All physical pictures including SEM or digital photos should be added a scale bar.

Author Response

Dear colleague,

Thank you for taking the time to review our publication and for your thorough analysis. We have carefully considered all of your questions and provided detailed responses below:

It is interesting that the PAN fibers containing Si given by two routes were compared completely. However, some issues need to be cleared or revised.

Question 1: The introduction of silicon source cannot result in obvious or special functionality of PAN fibers in the elucidation of Introduction part. And factually, this research showed this point that the strength of PAN fiber was not improved. Why do the authors introduce Silicon into PAN fibers? Or you should explain or evaluate further likely “novel” functions of as-spun PAN fibers you made.

Answer:

The main objective of this study was to demonstrate a novel method for in-situ introduction of particles into fibers via liquid droplets – solid particles transformation during the spinning process. TEOS was selected as the focus of our research due to its well-established properties and its potential to transform into solid particles during or after spinning. A corresponding phrase was added to the Manuscript.

An additional objective of our research was to produce carbon-silicon carbide fibers with improved resistance to air oxidation. We have successfully demonstrated the formation of silicon carbide within these fibers, thereby achieving our desired result.

Question 2: Based on the spinning processes, the decomposition of TEOS to form SiO2 is not easy. It’s well-known that the model process of casting film is very different from fiber forming process. The TEOS would be evaporated easily in the spinning process? The silicon of inner fibers identified by EDS is hard to be attributed to SiO2.

Answer:

Indeed, there are significant differences in the morphology of the particles formed in films and fibers, as depicted in Figure 4 and Figure 8, respectively. Since TEOS is a high-boiling liquid, it evaporates slowly. Furthermore, it undergoes rapid hydrolysis in the presence of water or ambient moisture. During the spinning process, TEOS undergoes hydrolysis/condensation leading to separation out of solution into a separate solid phase, particularly when exposed to air moisture. This hydrolysis process is most active during subsequent washing stages (in water) and thermal treatments. The resulting particles no longer evaporate but remain embedded within the fibers.

To analyze the PAN samples containing particles, we employed the EELS detector, known for its sensitivity to light elements and higher resolution of energy spectra. We compared the obtained data with various literature sources, including the Atlas of elemental compounds, where SiO- compounds have been extensively studied, especially in mineralogy. The acquired data, along with the presence of SiC indicated by X-ray analysis, strongly suggest a high likelihood of silica particle formation in the fibers.

Question 3: It is better if stress-strain curves were attached for Mechanical characteristics except for Table 5.

Answer:

 Indeed, full stress-strain curves are more informative. Unfortunately, the device used to measure the mechanical properties does not allow to record entire tensile load curves.

Question 4:  All physical pictures including SEM or digital photos should be added a scale bar.

Answer:

We checked the Manuscript again and scale bars were added to all Figures.

Round 2

Reviewer 2 Report

1.The foming of SiO2 was not explained clearly. Figure.4 showed the coating film and different from fiber, especially fiber interior. The SiO2 particles are different from those in fiber?
2.TEOS is not difficult to evaporate, and its boiling point, 169 °C even lower than that of DMSO (189 °C). 
3. "
SiO- compounds" exists TEOS as well, which cannot say it's SiO2. and "SiC indicated by X-ray analysis" showed the sample that treated under 1600 °C.

Author Response

Dear colleague, thank you again for your clarifying questions:

  1.  1.The foming of SiO2 was not explained clearly. Figure.4 showed the coating film and different from fiber, especially fiber interior. The SiO2 particles are different from those in fiber?

Answer: Indeed, the particles formed in the film are different from the particles in the fiber. We expect that similar processes occur in fibers obtained by this method, except for changes in particles size, as described in the text.
It was noticed the absence of thermal effects on DSC up to 600 degrees, indicates the predominant completion of hydrolytic polycondensation in the finished fiber.

  1. TEOS is not difficult to evaporate, and its boiling point, 169 °Ceven lower than that of DMSO (189 °C). 

Answer: The boiling point of TEOS is lower than that of DMSO, but it is actively hydrolyzed, followed by condensation and the formation of oligosiloxanes with significantly higher boiling points. So, for Ethyl silicate-40 (a product of HPC TEOS), the boiling point is already about 500 degrees. Thus, the oligosiloxanes formed from TEOS do not evaporate.

  1. "SiO- compounds" exists TEOS as well, which cannot say it's SiO2. and "SiC indicated by X-ray analysis" showed the sample that treated under 1600 °C.

Answer: I agree, we should choose the designations more correctly.